# OpenReview forum: "Leveraging Potential Violation Elements for LVLM-based Image Content Moderation"
_ICLR.cc/2026/Conference — ICLR 2026 Conference Withdrawn Submission_

### Official Review · Reviewer_P4xx · 2025-10-26

**Soundness:** 2
**Presentation:** 2
**Contribution:** 2
**Rating:** 4
**Confidence:** 4

**Summary:**

The paper introduces a new two-stage framework for image content moderation centered on PVEs as intermediate representations. The paper introduces PVE-100, a large-scale, manually annotated dataset with over 100 fine-grained elements. It also proposes a PVE perception objective that enables flexible two-stage fine-tuning by explicitly training models to recognize PVEs before policy-based classification. It also introduces a plug-and-play PVE perception expert that augments closed-source LVLMs (e.g., GPT-4o, Gemini-Pro) through explicit PVE-guided textual prompts without requiring model modification.

**Strengths:**

1. The paper presents a novel and well-motivated paradigm in content moderation, PVE, which decomposes image content moderation into interpretable, element-level perception and policy reasoning.
2. The proposed flexible two-stage finetuning with PVE perception objective significantly enhances LVLM’s content moderation performance.

**Weaknesses:**

1. The experiments only benchmark all the baselines on PVE-100, the authors’ own dataset. Although Table 4 briefly reports the proposed models’ results on LlavaGuard, the evaluation of other baselines is not included. Since PVE-100 shares a similar distribution with the training data, the comparison among models may not fully reflect the generalization ability. I suggest the authors carry out a more convincing, comprehensive evaluation on third-party benchmarks such as LlavaGuard, UnsafeBench, or MM-SafetyBench.
2. While the paper emphasizes flexibility, each safety policy (MM, MA, DY) still requires fine-tuning a separate model (Tables 1 & 2). This dependence on retraining undermines the claimed adaptability — if users wish to modify or update the policy, they must repeat the fine-tuning process. The approach, therefore, lacks true policy-agnostic inference or zero-shot generalization across policies.
3. The paper only reports precision, recall, F1-score, and accuracy. However, for moderation systems, false positive rate (FPR) and false negative rate (FNR) are critical indicators of practical safety performance. The absence of these metrics makes it difficult to assess trade-offs between over-blocking and under-blocking.

**Questions:**

1. How is the moderation task difficulty defined in the curriculum learning (CL) setup?
2. Could the authors report FPR/FNR for both baseline and proposed models? I believe these are key metrics in content moderation tasks.
3. Since each model is fine-tuned for a specific policy, can the authors provide results for policy transfer — for example, a model trained under Policy-MM but evaluated under Policy-DY?
4. Since the paper introduces a two-stage moderation process, what is the overall inference latency of the proposed models compared with baselines?
5. During the labeling process, how many annotators labeled each image? Were disagreements common across specific categories (e.g., ambiguous PVEs such as “suggestive gesture”)? How were these disagreements resolved to ensure annotation consistency?

---

### Official Review · Reviewer_VhEx · 2025-10-27

**Soundness:** 3
**Presentation:** 2
**Contribution:** 3
**Rating:** 6
**Confidence:** 4

**Summary:**

The authors introduce a new safety taxonomy consisting of 115 Potential Violation Elements (PVEs), which represent fine-grained safety categories for visual content moderation. Building on this taxonomy, they construct PVE-100, a dataset of 22,000 online images manually annotated according to the proposed PVEs. Using these annotations, the authors design different moderation policies by selecting specific subsets of PVEs and train and evaluate vision-language models (VLMs) under these policy settings. Their experiments empirically demonstrate the effectiveness of PVE-based training for improving VLM performance in content moderation tasks.

**Strengths:**

- The paper is overall well written and follows a logical and coherent narrative.

- PVE-100 appears to be a valuable contribution to the field of AI safety and visual content moderation.

- The authors report consistent performance improvements over baseline models, showing practical relevance for both open- and closed-source systems as well as the ability to generalize to new, unseen policies.

**Weaknesses:**

- In their work, the authors propose to separate perception and reasoning into two stages. First the model "perceives potential violation elements. It then evaluates these elements based on the understanding of policy guidelines and classifies the image into specific safety categories" (Fig 2). I find this framing not entirely convincing. In most VLM architectures, the vision encoder (e.g., CLIP) already handles perception by encoding visual information, while reasoning takes place within the language model component. If the CLIP encoder’s weights remain frozen during fine-tuning, the visual representations are not updated, which could also explain why the curriculum setting performed worse than joint training in their experiments.

- The authors introduce 115 PVE categories but later define three different policy subsets containing 7–11 categories each. It remains unclear how these subsets map onto the original taxonomy. Do they cover all safety dimensions or only a portion?

- Although the paper is generally well organized, the experimental setup lacks clarity. Do all your model setups generally require 2 passes through the model, or is this only the case for the proprietary models? What are the train/val data for the experiments? I assume table 1-6 all are trained on MM+MA+DY train set and evaluated on the respective val sets. Is the SFT model trained on safety rating only or does it also get the perception information?

**Questions:**

- If the SFT model is trained only on safety ratings, please compare it to a variant trained on both perception + rating, as this may help the model internalize reasoning behind decisions, an approach also adopted in most moderation systems (LlamaGuard, LlavaGuard, etc.).

- Consider including OpenAI’s moderation model as a baseline, as it is widely used and freely accessible.

- Since the training objectives are framed probabilistically, did you consider reporting or calibrating likelihoods/confidence scores for predicted perceptions and safety ratings?

- During tuning of the perception objective, is the vision encoder unfrozen? Do you think this objective could be fully integrated within the vision encoder rather than through the full VLM?

- The authors state: "Considering the generality, flexibility, and high performance of closed-source LVLMs, they remain a good choice for image content moderation." (292-294). However, such models are typically safety-aligned through fine-tuning. Did you observe alignment effects influencing model behavior? What happens if your moderation policy conflicts with the model provider’s alignment objectives? Can the model adapt or does it revert to its pre-aligned value system?


- How did the authors ensure that the 115 PVEs are of high quality, and how were the exact guidelines for evaluation and labeling defined? How are different severities within a single category handled and how do we draw a line between safe and unsafe? For instance, how would borderline cases (e.g., mild suggestive content versus explicit nudity) be distinguished and annotated?

- What is the source of the images in PVE-100? Are the generated captions used in your training setup? If not, do you think we could leverage them?


- In Table 6, if the results correspond to evaluation on the LLaVaGuard dataset, please state this explicitly in the caption.

- In Table 2, multiple results are highlighted in bold despite the note “the best performance is in bold.” Please revise to ensure consistent highlighting.

- Table 4 misses the base/untrained models.

**Details Of Ethics Concerns:**

The dataset involves online images with potentially sensitive or private content, yet the paper does not specify data provenance, consent, or compliance with content and privacy regulations. Although the authors describe a rigorous multi-stage annotation with professional annotators, there is no information about annotator protection, exposure mitigation, or compensation, which is critical when labeling potentially harmful or explicit material. An ethics review is recommended to ensure responsible data handling and fairness compliance.

---

### Official Review · Reviewer_4khR · 2025-10-30

**Soundness:** 3
**Presentation:** 4
**Contribution:** 2
**Rating:** 6
**Confidence:** 2

**Summary:**

This paper focuses on developing Large Vision-Language Models (LVLMs) for customized safety policies. To this end, the authors introduce PVE-100, the fine-grained, element-level dataset for visual content moderation. With element-level annotations, PVE-100 offers flexibility for evaluating and fine-tuning models. Their experimental results demonstrate that these fine-grained annotations can be leveraged to enhance open-source LVLMs and closed-source LVLMs via either fine-tuning or plug-and-play perception insertion.

**Strengths:**

1.	The motivation of this paper makes sense to me. LVLMs could benefit more from a fine-grained annotation of potential violation elements for images.

2.	This paper illustrates the applications of PVE-100. For both fine-tuning and inference, PVE-100 can help LVLMs make policy-based judgments about the visual contents.

**Weaknesses:**

1.	This paper contains content that may be offensive and disturbing in nature to readers, even though the authors have processed the images.

2.	It would be better to have a comprehensive comparison between the PVE-100 and existing datasets, which helps the readers identify the novelty of the proposed dataset clearly.

**Questions:**

1.	Please see the weaknesses No.2

2.	Where do you obtain the images of PVE-100 dataset?

**Details Of Ethics Concerns:**

I would like to bring this paper to the attention of the ethical community of ICLR. This paper contains content that may be offensive and disturbing in nature to readers, even though the authors have processed the images. In this case, I am concerned about the applied significance of this dataset.

---

### Official Review · Reviewer_3R8v · 2025-10-31

**Soundness:** 2
**Presentation:** 3
**Contribution:** 2
**Rating:** 2
**Confidence:** 4

**Summary:**

This paper introduces PVE-100, a fine-grained, element-level dataset and a corresponding two-stage framework to safeguard the output of generative image models: (1) perception of Potential Violation Elements and (2) policy-based judgment, where policy text determines whether a combination of elements constitutes a violation. Based on the introduced PVE perception objective for fine-tuning open-source models and an expert module that can guide closed-source models via explicit textual hints, the authors investigate different training strategies. Experiments demonstrate improved performance and adaptability across multiple community policy scenarios.

**Strengths:**

- PVE-100 provides a large-scale, fine-grained annotation scheme decomposing unsafe content into entities, actions, and contexts, enabling policy-adaptive moderation.
- The methodology is conceptually sound, with clear motivation. The two-stage factorization is intuitive and interpretable.
- The paper is generally clear and well organized, and the appendices are thorough
- The empirical validation appears to be largely solid, showing consistent improvements across several LVLM families and multiple real-world policies.
- Results seem to be reproducible. The authors provide details in the appendix and provide a release plan with appropriate safeguards.

**Weaknesses:**

Summarized, while methodology is conceptually sound, with clear motivation and solid empirical validation, several modeling and evaluation aspects remain unclear:

- The elements of the proposed taxonomy and the policies used are only used to define what is unsafe, not as exceptions to consider something safe. For example, a gun carried by a soldier/policeman could be considered safe, while a gun carried by other persons is considered unsafe. In contrast, previous work, such as Helff et al. (2025, ICLR), make use of context exceptions or “can”-conditions.
- Using an LVLM to read textual policy rules introduces unnecessary ambiguity. Instead, a symbolic or first-order logic layer grounded in the detected PVEs could offer more transparent reasoning. The choice of LVLM usage in this case is unclear.
- The experiments do not report cross-validated results. It is unclear whether the gains reported in Tables 2 and 3 are robust.
- The relationship between the proposed PVE taxonomy and existing safety taxonomies (e.g., Zeng et al., 2024; Helff et al., 2025) is not well-articulated. E.g. what is used in the present taxonomy and what is discarded and why?

## Ethical concern

The dataset description raises a significant ethical and legal concern.
In Appendix C.3 (“Safety Scale”), the authors explicitly list child nudity and child abuse among the Severe categories of annotated visual elements. However, the paper does not clarify:
- whether any actual images containing such material were collected or retained,
- how the dataset acquisition process ensured compliance with laws on child sexual abuse material (CSAM), and
- whether any filtering, synthetic replacement, or redaction was performed before annotation.


## Minor:
- Figure 1 font size of center part could be increased. Further it is unclear why some samples lack one or more of the three semantic types.
- ShieldGemma 2 as baseline missing (Zeng et al. 2025, ShieldGemma 2: Robust and Tractable Image Content Moderation)


## References:

Zeng et al. AI risk categorization decoded (AIR 2024): From government regulations to corporate policies, 2024

Helff et al. LlavaGuard: An Open VLM-based Framework for Safeguarding Vision Datasets and Models, 2025

**Questions:**

1. Could you elaborate on how context influences the moderation outcome? Why have you decided to only use „should not“ cases?
2. Which and how were “real-world community policies” selected (line 90)?
3. Why rely on textual policy prompts interpreted by the LVLM instead of a rule-based engine over the PVE space?
4. Have you considered adapter-based modular training (e.g., LoRA adapters) for perception vs. policy reasoning, instead of joint or curriculum setups?

**Details Of Ethics Concerns:**

The dataset description raises a significant ethical and legal concern.
In Appendix C.3 (“Safety Scale”), the authors explicitly list child nudity and child abuse among the Severe categories of annotated visual elements.
However, the paper does not clarify:
- whether any actual images containing such material were collected or retained,
- how the dataset acquisition process ensured compliance with laws on child sexual abuse material (CSAM), and
- whether any filtering, synthetic replacement, or redaction was performed before annotation.

Even if the dataset only uses blurred or simulated representations, the current text could be interpreted as indicating that real child-sexual-abuse imagery was handled by annotators. Possession, annotation, or redistribution of such material, even for research, is illegal in most jurisdictions and subject to strict reporting requirements.

Because the paper commits to publicly releasing the dataset (“available upon acceptance”), these omissions warrant urgent clarification and likely an ethics review to ensure that:
1. no illegal material was collected or retained,
2. any sensitive content was simulated or procedurally generated, and
3. the dataset release process conforms to international CSAM and data-protection regulations.

---

### Note · Authors · 2025-11-12

I have read and agree with the venue's withdrawal policy on behalf of myself and my co-authors.